# Adolescent suicide attempts in Brazil and impact of COVID-19 pandemic: A temporal analysis

Juliana D. P. Bulhões[1,2,3�], Gabriela L. Rosier[1,2,3,4�], Klauss Villalva-Serra[2,3,4], Fabio M. H. Sales Filho[5], Moreno M. Magalhães[5], Beatriz Barreto-Duarte[1,2,4‡], Bruno B. Andrade[1,2,3,4‡]*

1 Escola Bahiana Medicina e Saúde Pública, Salvador, Bahia, Brazil, 2 Laboratório de Pesquisa Clínica e Translacional, Instituto Gonçalo Moniz, Fundação Oswaldo Cruz, Salvador, Bahia, Brazil, 3 Multinational Organization Network Sponsoring Translational and Epidemiological Research (MONSTER) Initiative, Salvador, Bahia, Brazil, 4 Institute for Research in Priority Populations (IRPP), Multinational Organization Network Sponsoring Translational and Epidemiological Research (MONSTER) Initiative, Salvador, Brazil, 5 Laboratório de Análise e Visualização de Dados, Fundação Oswaldo Cruz, Porto Velho, Brazil

These authors are contributed equally to this work.
‡ BB-D and BBA also contributed equally to this work.
* bruno.andrade@fiocruz.br

## Abstract

Suicide is a leading cause of adolescent mortality worldwide. Exogenous poisoning— defined as exposure to toxic substances such as medications or chemicals— is a common method, but large-scale epidemiological data in Brazil are limited. This study describes sociodemographic profiles, temporal trends, regional disparities, and the COVID-19 pandemic's impact on adolescent suicide attempts by exogenous poisoning in Brazil from 2018 to 2023. We conducted a study, the SINAN database, on reported suicide attempts by exogenous poisoning in adolescents aged 10–19 years nationwide. Descriptive analyses were performed by sex, age group, toxic agent, and care setting. Temporal trends were evaluated with LOESS decomposition, the Mann–Kendall trend test, and interrupted time-series analysis (ITSA) assessed the impact of COVID-19 on suicide attempt rates. A total of 142 251 attempts were identified; notification was substantially more common among the 15–19 age group (75.4%) and females in both groups. Medications accounted for most reported agents, and most events occurred at home. Nationwide incidence rose steadily from 2018, peaked in 2019, declined significantly in 2020 (p < 0.01), and resumed increasing through 2022. Upward trends were observed in all regions, with the 10–14-year group exhibiting the largest relative increase. The Southeast region maintained the highest cumulative incidence. ITSA indicated underreporting during the pandemic, with a 66.5% decrease in ages 10–14 and a 56.2% drop in ages 15–19. The high and rising incidence of adolescent suicide attempts by exogenous poisoning in Brazil highlights the need for strengthened mental health interventions, enhanced pharmacovigilance, and targeted prevention, especially post-pandemic.

**Data availability statement:** All data underlying the findings are derived from the publicly available SINAN database (Brazilian Information System for Notifiable Diseases), accessible at: https://datasus.saude.gov.br/informacoes-de-saude-tabnet/.

**Funding:** The study was supported by the Intramural Research Program of the Fundação Oswaldo Cruz (B.B.A.). B.B.D received a fellowship from Coordenação de Aperfeiçoamento de Pessoal de Nível Superior (Finance code: 001). B.B.A, is a senior investigators and fellows from the Conselho Nacional de Desenvolvimento Científico e Tecnológico (CNPq), Brazil. The funders had no role in study design, data collection and analysis, decision to publish, or preparation of the manuscript.

**Competing interests:** The authors have declared that no competing interests exist.

## Introduction

In 2019, suicide was responsible for one death every 40 seconds worldwide, representing a significant global public health challenge [1]. Suicide is defined as a voluntary act in which an individual intentionally takes their own life. Prior to a fatal suicide, an individual typically undergoes multiple suicide attempts—on average, 20 per completed suicide [2]. Suicide attempts are characterized as self-inflicted, potentially harmful behaviors that do not result in death but pose serious health risks [3]. Among the various methods of suicide, exogenous poisoning stands out as the leading method of suicide attempts and ranks among the top three methods of suicide worldwide [4–6]. Both suicide and suicide attempts are complex and multifactorial phenomena, influenced by a range of factors, including gender, geographic region, socioeconomic conditions, and sociopolitical contexts [1,3,4].

Adolescents and young adults are particularly vulnerable to suicidal behavior. Suicide is one of the three leading causes of death among individuals aged 15–29 years. Suicidal behavior in this age group is driven by a combination of biological, psychological, and social factors, including depressive mood, substance abuse, family conflicts, childhood trauma (such as physical and sexual abuse), neglect, and a family history of psychiatric disorders [3,7,8]. These risk factors are exacerbated in low- and middle-income countries (LMICs), where limited access to mental health care, social inequalities, and economic instability further increase the risk of suicide.

In Brazil, a country marked by significant socioeconomic disparities and a rapidly growing adolescent population, the suicide rate among individuals aged 15–24 years was 7.6 per 100,000 inhabitants [1]. This public health concern has been further aggravated by increased antidepressant use, rising rates of depression, and prolonged social isolation during the COVID-19 pandemic.

Despite the increasing burden of suicide attempts by exogenous poisoning, there is a lack of national studies exploring this phenomenon in Brazil. Understanding the socioepidemiological characteristics and temporal trends of suicide attempts is crucial to developing targeted prevention strategies and evidence-based public health policies. Therefore, this study aims to characterize the socioepidemiological profile of suicide attempts by exogenous poisoning among Brazilian adolescents from 2018 to 2023. Additionally, we seek to analyze temporal trends and the geographic distribution of notified cases, providing insights into regional disparities and patterns across the country.

## Materials and methods

### Study design, population, and data collection

This is a nationwide ecological study conducted in Brazil, focusing on cases of suicide attempts via exogenous poisoning among adolescents aged 10–19 years between 2018 and 2023. The data were obtained from the database of the Information System for Notifiable Diseases (SINAN).

Brazil is a middle-income country with a universal public health system, the *Sistema Único de Saúde* (SUS), which provides free access to health services at all

levels of care. Mental health care for children and adolescents is structured through the *Rede de Atenção Psicossocial* (RAPS), integrating primary, secondary, and tertiary services. Within RAPS, the *Centros de Atenção Psicossocial Infantil* (CAPS I) are specialized centers that provide care for children and adolescents with severe or persistent psychological distress — including substance-related issues — and are recommended for municipalities with populations over 70,000 inhabitants. These centers play a key role in early identification, crisis management, and longitudinal mental health care for youth.

Brazil also has robust child protection and human rights legislation, including the *Estatuto da Criança e do Adolescente* (Child and Adolescent Statute), ensuring the right to health, education, and protection from violence. However, regional inequalities, political instability, and socioeconomic disparities still challenge the consistent implementation of such programs nationwide.

The study population consisted of suicide attempt cases extracted from SINAN's database of exogenous poisonings. Inclusion criteria included individuals aged 10–19 years diagnosed with exogenous poisoning, where the exposure circumstance was classified as a suicide attempt (S1 Fig).

The collected variables included: age (Categorized as 10–14 years and 15–19 years), sex (Female, male, or unknown), race (White, Black, mixed-race, Asian, Indigenous, or unknown), location of exposure (home, workplace, commuting to work, healthcare facility, school/daycare, or external environment), year of occurrence, exposure circumstance (suicide attempt), route of exposure (digestive, cutaneous, respiratory, ocular, transplacental, vaginal, or unknown), type of exposure (acute single, acute repeated, acute on chronic, chronic, or unknown), type of care (outpatient, home, hospital, or unknown), toxic agent group (medications, agricultural pesticides, domestic pesticides, public health pesticides, food and beverages, cosmetics and personal hygiene products, drugs of abuse, metals, toxic plants, veterinary products, household products, industrial chemicals, rodenticides, or unknown).The study aimed to describe these variables and analyze regional trends.

## Statistical analysis

Data processing and statistical analyses were conducted using R and Python. Descriptive statistics were performed to summarize the dataset, with categorical variables expressed as absolute and relative frequencies (n, %) and continuous variables reported as medians and interquartile ranges (IQR). The normality of continuous variables was assessed using the Shapiro-Wilk test, and non-parametric tests were used where appropriate.

To evaluate temporal trends, a time series decomposition analysis was performed, separating the observed data into trend, seasonality, and residual components using the Seasonal-Trend Decomposition based on LOESS (STL decomposition). This method enables the identification of long-term patterns and cyclical variations, controlling for seasonal effects. The Mann-Kendall test, a non-parametric test for detecting monotonic trends in time series data, was also applied to assess whether there was a statistically significant increase or decrease in the number of suicide attempt notifications over time. The test was stratified by age group (10–14 and 15–19 years) and Brazilian region to capture differences across subpopulations.

For regional analysis, the cumulative incidence per 100,000 inhabitants was calculated for each Brazilian region, using official population estimates from the Brazilian Institute of Geography and Statistics (IBGE) Census. Comparisons between categorical variables were performed using the Chi-square test or Fisher's exact test, while comparisons between continuous variables were conducted using the Mann-Whitney U test due to the non-normal distribution of the data.

Finally, aiming to evaluate how the COVID-19 pandemic impacted the notification of suicide attempts via exogenous poisoning, at a regional and national level, we performed an interrupted time-series analysis (ITSA). We used monthly notification data, with the interruption point being set to April 2020, when the COVID-19 pandemic began in Brazil. Data up until the intervention point was then fitted using an Integrated Nested Laplace Approximation model to forecast

counterfactual data monthly notification, in a scenario without the pandemic, until December 2023. Then absolute and relative differences were calculated comparing the observed data on pandemic notification to the hypothetical counterfactual evaluations.

All statistical tests were two-tailed, with a significance level set at p < 0.05. Analyses were performed using the statsmodels, NumPy, and SciPy libraries in Python, while for R the base stats and R-INLA statistical packages were used.

### Ethical considerations

This study utilized data from the SINAN database for the period between 2018 and 2023. As the study relied on a governmental platform that is publicly available (https://datasus.saude.gov.br/informacoes-de-saude-tabnet/), anonymized, and pre-processed by the Brazilian Ministry of Health—including verification of duplicate records—it adhered to the ethical standards outlined in Resolution No. 466/12 of the National Health Council regarding research ethics. Therefore, the study was exempt from approval by an Ethics Committee (CEP).

## Results

### Characterization of Brazilian adolescents in suicide attempts

Between 2006 and 2023, a total of 1,875,442 cases of exogenous poisoning were reported in Brazil. After applying the inclusion criteria—age (10–19 years), study period (2018–2023), and exposure classified as a suicide attempt—a total of 142,251 cases were analyzed. These cases were stratified into two age groups: 10–14 years (n = 34,888) and 15–19 years (n = 107,363). In both age groups, females were the most affected, comprising 89.6% and 78.4% of cases, respectively. The most frequently reported racial group was White (45.9% in 10–14 years and 45.7% in 15–19 years), followed by mixed-race individuals (38.6% and 38.4%).

Furthermore, the majority of exposures were classified as an acute single exposure (68.3% and 64.3%), and most events occurred within the case's residency (89.4% and 89.5%). Regarding mode of exposure, almost all cases were exposed through the digestive route (95.6%), with medications being the most predominant toxic agents in both age groups (87% in the 10–14 age group and 84.4% in the 15–19 age group). Finally, hospital care was required in 76.8% of notified cases aged between 10–14 years old and 76.6% in the 15–19 age group (Table 1).

### Number of suicide attempts by exogenous poisoning

Throughout the study period, the highest number of suicide attempt notifications was consistently observed among adolescents aged 15–19 years. Both age groups followed a similar seasonal pattern, with a peak in notifications in 2019 (20,833 cases), followed by a decline in 2020 (14,872 cases), coinciding with the COVID-19 pandemic. A slow increase was observed in 2021, with an acceleration in cases from 2022 onwards. In 2023, the number of cases reached 21,077, with a more pronounced increase in the 15–19 age group (Fig 2).

### Regional distribution of suicide attempts in Brazil

The geographic distribution of suicide attempts by state and region was also analyzed. The Southeast recorded the highest number of cases, totaling 66,607 notifications (46.8% of all national cases), with 16,199 cases (46.4%) in the 10–14 age group and 50,408 cases (47%) in the 15–19 age group. The South region reported 30,700 cases (21.6%), with 7,848 cases (22.5%) in the 10–14 age group and 22,852 cases (21.3%) in the 15–19 age group. The Northeast had 26,483 cases (18.6%), with 6,459 cases in the 10–14 age group and 20,024 cases in the 15–19 age group. In the Central-West, a total of 13,519 cases (9.5%) were recorded, with 3,266 cases in the 10–14 age group and 10,253 cases in the 15–19 age group. The North region had the lowest number of notifications, accounting for 3.5% of the total cases.

Additionally, we also calculated cumulative incidence per 100,000 inhabitants for each region. Among adolescents aged 10–14 years, the South region showed the highest incidence, followed by the Southeast and Central-West. When

**Table 1. Epidemiological characteristics of Brazilian Adolescents victims of suicide attempts according to age group.**

| Characteristics | All (n = 142,251) | 10-14 years old (n = 34,888) | 15-19 years old (n = 107,363) |
|---|---|---|---|
| **Sex, n (%)** | | | |
| Female | 115,439 (81.2) | 31,258 (89.6) | 84,181 (78.4) |
| **Race, n (%)** | | | |
| White | 65,225 (45.9) | 15,941 (45.7) | 49284 (45,9) |
| Mixed | 54,745 (38.5) | 13,481 (38.6) | 41264 (38,4) |
| Black | 6,988 (4.9) | 1,602 (4.6) | 5,386 (5.0) |
| Yellow | 960 (0.7) | 235 (0.7) | 725 (0,7) |
| Indigeneous | 384 (0.3) | 86 (0.2) | 298 (0,3) |
| Ignored | 13,236 (9.3) | 3,392 (9.7) | 9844 (9 2) |
| Missing | 713 (0.5) | 151 (0.4) | 562 (0,5) |
| **Exposure type, n (%)** | | | |
| Acute - single | 92,836 (65.3) | 23,825 (68.3) | 69,011 (64.3) |
| Acute - repeated | 26,957 (19.0) | 6,027 (17.3) | 20,930 (19.5) |
| Acute under chronic | 1,032 (0.7) | 209 (0.6) | 823 (0.8) |
| Chronic | 959 (0.7) | 199 (0.6) | 760 (0.7) |
| Missing | 20,467 (14.4) | 4,628 (13.2) | 15,839 (14.8) |
| **Location of occurrence, n (%)** | | | |
| Residence | 127,313 (89.5) | 31,340 (89.8) | 95,973 (89.4) |
| School/day care center | 2,261 (1.6) | 1,075 (3.1) | 1,186 (1.1) |
| Others | 1,900 (1.3) | 409 (1.2) | 1,491 (1.4) |
| External environment | 1,005 (0.7) | 161 (0.5) | 844 (0.8) |
| Workplace | 473 (0.3) | 33 (0.1) | 440 (0.4) |
| Health services | 143 (0.1) | 36 (0.1) | 107 (0.1) |
| Work commute | 37 (0.0) | 3 (0.0) | 34 (0.0) |
| Ignored | 7,431 (5.2) | 1,406 (4.0) | 6,025 (5.6) |
| Missing | 1,688 (1.2) | 425 (1.2) | 1,263 (1.2) |
| **Exposure pathway, n (%)** | | | |
| Digestive tract | 135,925 (95.6) | 33,320 (95.5) | 102,605 (95.6) |
| Respiratory tract | 346 (0.2) | 58 (0.2) | 288 (0.3) |
| Parenteral | 103 (0.1) | 21 (0.1) | 82 (0.1) |
| Cutaneous | 146 (0.1) | 32 (0.1) | 114 (0.1) |
| Transplacental | 8 (0.0) | 1 (0.0) | 7 (0.0) |
| Vaginal route | 3 (0.0) | 2 (0.0) | 1 (0.0) |
| Eyepiece | 21 (0.0) | 4 (0.0) | 17 (0.0) |
| Others | 213 (0.1) | 48 (0.1) | 165 (0.2) |
| Ignored | 520 (0.4) | 113 (0.3) | 407 (0.4) |
| Missing | 4,966 (3.5) | 1,289 (3.7) | 3,677 (3.4) |
| **Groups of toxic agents, n (%)** | | | |
| Medications | 121,034 (85.1) | 30,367 (87.0) | 90,667 (84.4) |
| Raticides | 3,607 (2.5) | 552 (1.6) | 3,055 (2.8) |
| Domestic cleaning products | 3,424 (2.4) | 826 (2.4) | 2,598 (2.4) |
| Pesticide agents for agricultural use | 1,282 (0.9) | 219 (0.6) | 1,063 (1.0) |
| Abused drugs | 1,010 (0.7) | 157 (0.5) | 853 (0.8) |
| Others | 1,030 (0.7) | 196 (0.6) | 834 (0.8) |

*(Continued)*

**Table 1.** (Continued)

| Characteristics | All (n = 142,251) | 10-14 years old (n = 34,888) | 15-19 years old (n = 107,363) |
|---|---|---|---|
| Pesticide agents for household use | 754 (0.5) | 162 (0.5) | 592 (0.6) |
| Veterinary product | 597 (0.4) | 124 (0.4) | 473 (0.4) |
| Industrial chemical products | 630 (0.4) | 130 (0.4) | 500 (0.5) |
| Cosmetics/ personal hygiene product | 319 (0.2) | 76 (0.2) | 243 (0.2) |
| Food and drink | 174 (0.1) | 36 (0.1) | 138 (0.1) |
| Toxic plant | 80 (0.1) | 19 (0.1) | 61 (0.1) |
| Metal | 177 (0.1) | 31 (0.1) | 146 (0.1) |
| Pesticide agents for public health use | 40 (0.0) | 11 (0.0) | 29 (0.0) |
| Ignored | 986 (0.7) | 232 (0.7) | 754 (0.7) |
| Missing | 7,107 (5.0) | 1,750 (5.0) | 5,357 (5.0) |
| **Type of care, n (%)** | | | |
| Hospital | 108,996 (76.6) | 26,805 (76.8) | 82,191 (76.6) |
| Outpatient | 29,820 (21.0) | 7,232 (20.7) | 22,588 (21.0) |
| Homecare | 584 (0.4) | 149 (0.4) | 435 (0.4) |
| None | 373 (0.3) | 123 (0.4) | 250 (0.2) |
| Ignored | 1,180 (0.8) | 274 (0.8) | 906 (0.8) |
| Missing | 1,298 (0.9) | 305 (0.9) | 993 (0.9) |
| **Region, n (%)** | | | |
| Midwest | 13519 (9.5) | 3266 (9.4) | 10253 (9.5) |
| Northeast | 26483 (18.6) | 6459 (18.5) | 20024 (18.7) |
| North | 4934 (3.5) | 1116 (3.2) | 3818 (3.6) |
| Southeast | 66607 (46.8) | 16199 (46.4) | 50408 (47.0) |
| South | 30700 (21.6) | 7848 (22.5) | 22852 (21.3) |
| Missing | 8 (0.0) | – | 8 (0.0) |

Note: Data represent no. (%). Patients were divided into those aged from 10-14 years and 15–19 years.

comparing the 15–19 age group, a higher cumulative incidence was observed among the older group in all regions, nonetheless following a similar distribution to the younger age group, except for elevated rates in the Central-West (Fig 1).

### Trend analysis by region and age group

A temporal trend analysis revealed a significant increase in the number of suicide attempt notifications across both age groups in most regions. For most regions, including Southeast, Central-West, the North and Northeast, significant positive trends were found in both age groups (p < 0.0001 for 10–14 age group & p < 0.0001 for 15–19 age group). However, in the South region this significant increasing trend was only identified in the in the 10–14 age group (p < 0.0001), while no significant trend was observed for the 15–19 age group (p < 0.0632) (S2 Fig).

### Impact of COVID-19 pandemic on notifications

An interrupted time-series analysis was conducted using monthly suicide attempt notifications to evaluate the impact of the COVID-19 pandemic on surveillance at both national and regional levels. As shown in S1 Table and Fig 2, significant reductions in reported attempts were observed across both age groups in 2020, with a 66.5% decrease among individuals aged 10–14 and a 56.2% decline in those aged 15–19.

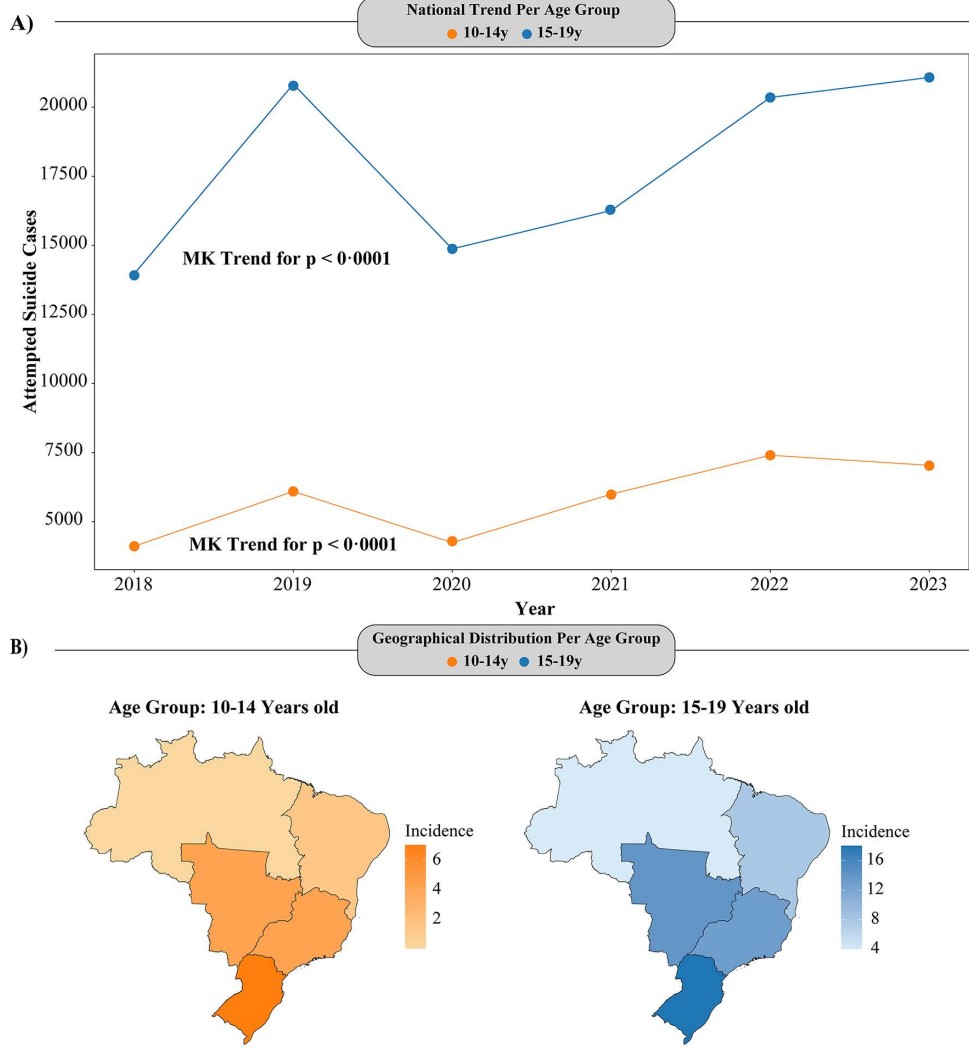

**Fig 1. Cumulative incidence of suicide attempts by exogenous poisoning per 100,000 inhabitants, stratified by age groups (10–14 and 15–19 years) (A) and Brazilian regions (B), 2018–2023.**

At the sub-national level, the poorest regions of Brazil, particularly the North and Northeast, experienced the most significant declines in reported suicide attempts across both age groups.

In the North, observed suicide attempt cases among 10- to 14-year-olds dropped by -75.9% ([-39.8% to -112.1%]) compared to counterfactual estimates, while the 15- to 19-year-old group saw a -77.4% decline ([-64.8% to -90.8%]). Similarly, in the Northeast, there was a -51.4% reduction ([-34.3% to -68.4%]) among 10- to 14-year-olds and a -64.4% decrease ([-54.4% to -74.3%]) in 15- to 19-year-olds (S2 Table, Figs 3 and 4).

However, our models also indicate that underreporting trends normalized by late 2021, with observed data from 2022 onward aligning with or even surpassing counterfactual estimates, suggesting a potential rebound in notification rates.

## Discussion

Suicide is a critical global public health problem, with high incidence rates among adolescents and young adults [1]. Understanding the socio-epidemiological characteristics of populations at risk of suicide attempts is essential for

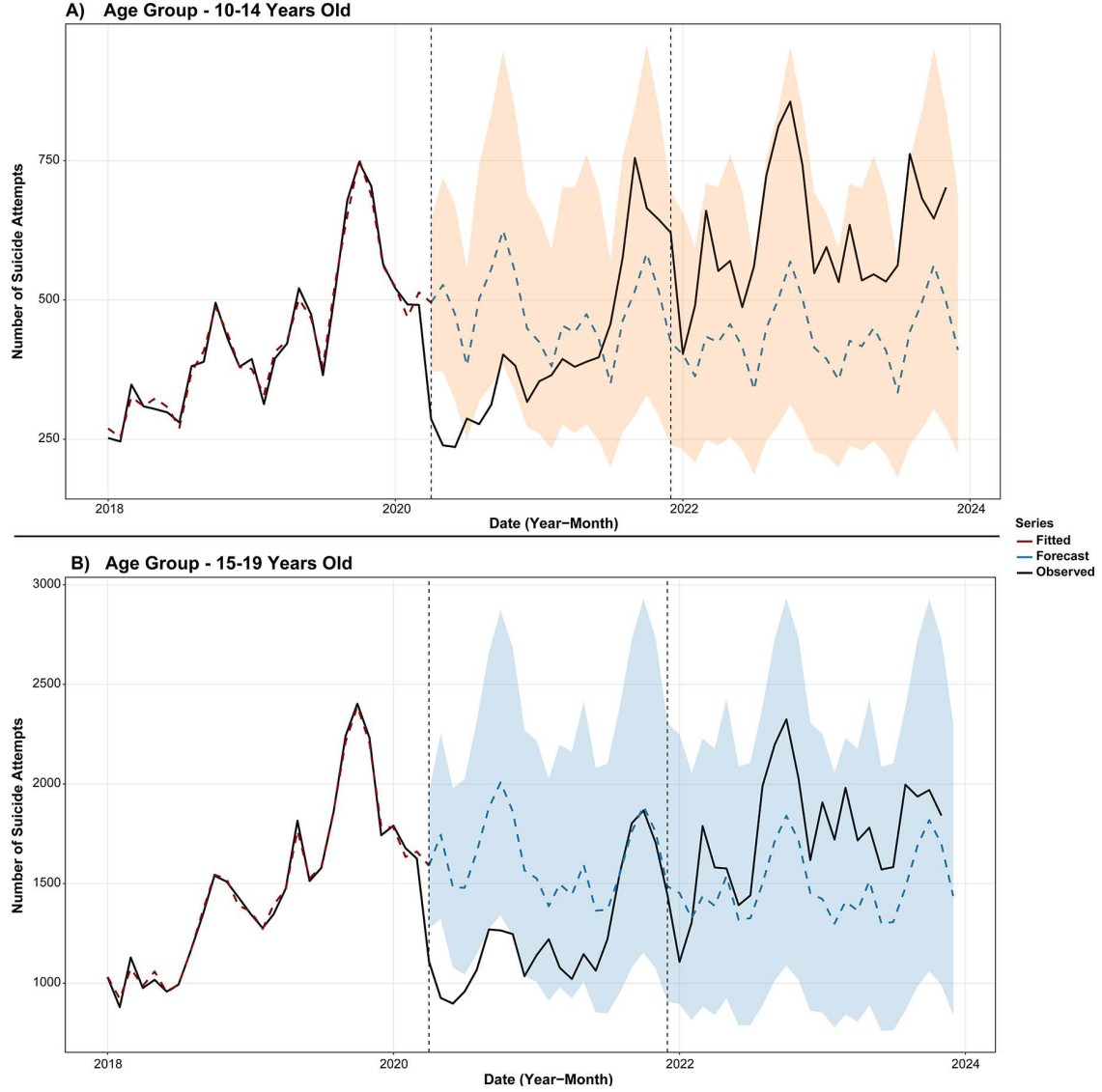

**Fig 2. National impact of the COVID-19 pandemic on suicide attempt notifications by exogenous poisoning, stratified by age groups (10–14, A, and 15–19 years, B).**

developing effective prevention policies, particularly in low- and middle-income countries (LMICs) such as Brazil, where multiple layers of social vulnerability compound the existing risk factors faced by this population. This study provides a comprehensive analysis of suicide attempts by exogenous poisoning among Brazilian adolescents between 2018 and 2023, revealing important epidemiological trends.

The findings indicate a higher prevalence among females, a pattern consistent with global trends that suggest women are more likely to engage in non-lethal self-harm behaviors, whereas men tend to choose more lethal methods, leading to higher mortality rates [9]. Studies have shown that male suicide success rates are two to five times higher than those of females, likely due to the use of more violent means, such as firearms or hanging [10]. In the United States, between 1975 and 2016, suicide rates among female adolescents, particularly those aged 10–14 years, increased significantly, whereas

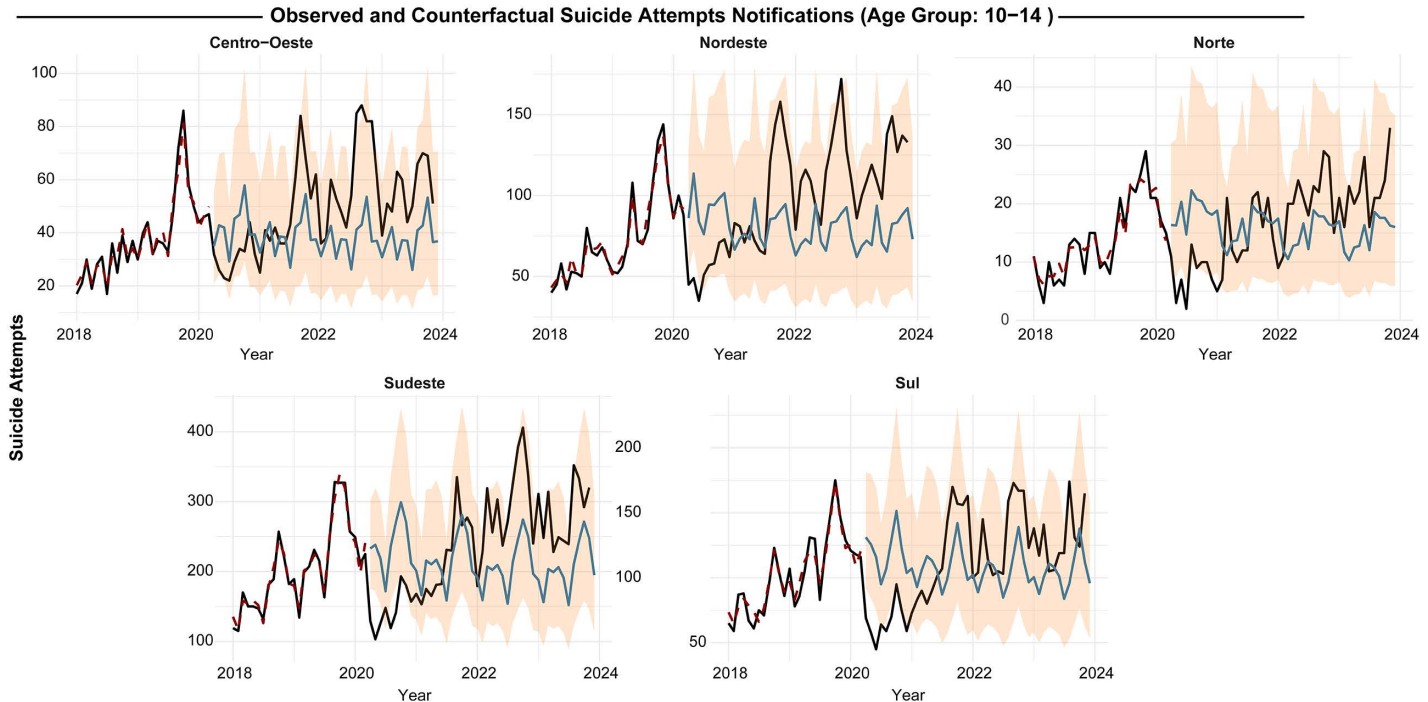

**Fig 3. Regional impact of the COVID-19 pandemic on suicide attempt notifications by exogenous poisoning at those aged 10–14 years old.**

rates among male adolescents declined [11]. These trends have been attributed to a higher frequency of previous suicide attempts among females and changes in the lethality of methods used [12]. Consistently, our study identified a higher incidence of suicide attempts among Brazilian female adolescents, predominantly involving medications, with most events occurring at home. These findings are in line with studies conducted in France and Brazil, which highlight the significant role of medication accessibility in adolescent suicide attempts [13]. Further research is needed to assess the specific drug classes involved, pathways of access, and the role of family supervision in preventing such incidents.

Between 2018 and 2023, an increase in suicide attempt notifications was observed across all age groups and regions, with notable rises in the South and Southeast regions. This pattern aligns with global trends, which highlight rising suicide rates in South America, with Brazil being a significant contributor to this surge [12]. The study also identified a sharp increase in cases in 2019, followed by a significant decline in 2020, coinciding with the COVID-19 pandemic This temporary reduction likely reflects disruptions in healthcare services, reduced healthcare-seeking behavior, and lower case reporting due to lockdown restrictions, rather than a true decrease in suicidal behavior. During this period, adolescents had less contact with schools, which are important sources of suicide attempt notifications, while exposure to domestic stressors increased and peer interactions declined. Although corporal punishment and peer violence at schools may have temporarily decreased, emotional distress, online exposure, and household conflicts intensified, creating a complex risk environment. The subsequent rise in 2021 and acceleration through 2022 and 2023 suggest a delayed but significant impact of the pandemic on adolescent mental health [13,14].

The regional distribution of suicide attempts revealed higher incidence rates in the South, Southeast, and Central-West regions, while the North exhibited the lowest rates. These findings align with a systematic review of adolescent suicides in Brazil, which identified Porto Alegre (South) and Belo Horizonte (Southeast) as having the highest adolescent suicide rates [15]. The disparities observed in this study may reflect differences in healthcare infrastructure, socioeconomic conditions, and access to mental health services across Brazilian regions. Regions with better healthcare coverage may exhibit

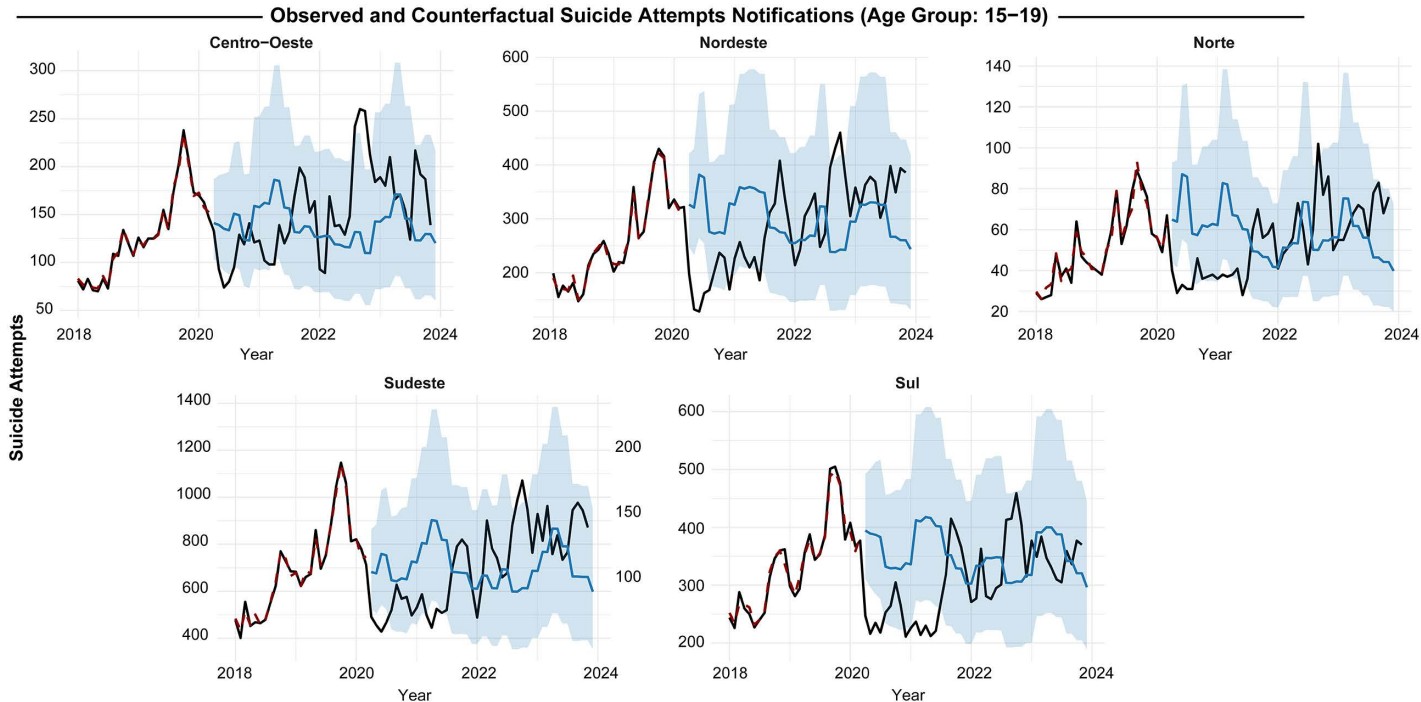

**Fig 4. Regional impact of the COVID-19 pandemic on suicide attempt notifications by exogenous poisoning at those aged 15–19 years old.**

higher recorded rates due to more comprehensive reporting mechanisms, whereas underreporting in regions with limited healthcare access may lead to artificially low suicide attempt rates [16].

The Central-West region showed a notably high incidence of suicide attempts in the 15–19 age group, raising concerns about underlying social determinants, including economic instability, migration patterns, and barriers to healthcare access. The time series decomposition and trend analysis indicated a significant upward trend in suicide attempts across most regions, particularly among the 10–14 age group in the South, which is consistent with recent national evidence indicating rising suicide rates among younger adolescents in Brazil over the past two decades [17]. While younger adolescents (10–14 years) are more susceptible to family and school-related stressors, older adolescents face additional pressures such as identity formation, romantic relationships, and social media exposure. Moreover, increased screen time, reduced peer interaction, and academic instability during the pandemic may have further amplified mental health risks.

The seasonal fluctuations observed in suicide attempts emphasize the importance of identifying high-risk periods, which may be linked to academic stress, social isolation, or economic crises. Although specific studies on seasonal patterns in adolescent suicide attempts in Brazil are limited, international research has shown higher rates of suicidal behavior during specific times of the year, often associated with academic examination periods and social transitions [18].

The increased demand for hospital care due to suicide attempts highlights the burden on the healthcare system and reinforces the need for mental health screening in emergency and primary care settings. A cross-sectional study on suicide ideation and attempts among adolescents emphasized the critical role of healthcare services in early identification and intervention, advocating for the integration of mental health assessments into routine medical care [19]. Additionally, the predominance of suicide attempts occurring at home reinforces the need for family-centered prevention strategies, educational interventions, and stricter control of pharmaceutical access. Previous studies have emphasized that adolescents often use readily available substances for suicide attempts, suggesting that limiting access to potential means within the home environment could be an effective preventive measure [20]. A key strength of this study is its nationwide

scope and large sample size, which allows for a robust epidemiological analysis across different regions and demographic groups. The use of official government data (SINAN) enhances the reliability of findings, ensuring that data collection follows standardized national reporting protocols.

This study has some limitations. Underreporting and misclassification of cases remain concerns, particularly in under-resourced regions where healthcare access is limited. The low notification rates in the North suggest possible underreporting, highlighting the need for further studies to clarify regional discrepancies. The absence of detailed information on psychiatric comorbidities, family history, and previous suicide attempts restricts the ability to assess individual risk factors comprehensively. In addition, the SINAN database does not allow for a detailed distinction between early and late adolescence, which limits age-specific analyses and interpretation of developmental risk patterns. Furthermore, the database does not include variables related to domestic or sexual violence, early marriage, or teenage pregnancy, which limits the ability to explore gender-related determinants of suicidal behavior. These omissions highlight the need for future studies to integrate social and gender-based indicators into national surveillance systems. Missing data for certain variables was notably high, and the open nature of the medication notification field complicates the precise identification of drug classes, increasing the likelihood of errors or omissions. Future research should focus on qualitative investigations to explore adolescents' motivations, social determinants, and family dynamics contributing to suicidal behavior. Additionally, the application of machine learning techniques and predictive modeling could help identify high-risk groups and improve targeted prevention strategies.

This study provides critical insights into the epidemiology of suicide attempts by exogenous poisoning among Brazilian adolescents, highlighting regional disparities, emerging trends in younger age groups, and the lasting impact of the COVID-19 pandemic on adolescent mental health. The findings underscore the urgent need for strengthened public health policies, improved mental health infrastructure, and community-based interventions to address the rising burden of adolescent suicidal behavior.

By implementing evidence-based prevention strategies within Brazil's existing public health and education systems, suicide prevention can be more effectively integrated into primary care and school-based mental health initiatives. Strengthening collaboration between the health and education sectors, promoting anti-bullying programs, and ensuring early identification of psychological distress are essential. Developing a coordinated national pathway for adolescent mental health care would enhance intersectoral prevention efforts and ensure continuity of support. These measures are consistent with Brazil's national mental health and education policies and reflect the country's commitment to advancing adolescent well-being. Strengthening pharmacovigilance and improving access to mental health services are also crucial to reducing suicide attempt rates and mitigating long-term consequences for at-risk youth.

## Supporting information

**S1 Fig. Flowchart of the study population, illustrating the selection process based on inclusion for suicide attempts by exogenous poisoning in Brazil, 2018–2023.**
(TIF)

**S2 Fig. Time series decomposition with trend analysis of suicide attempt notifications by exogenous poisoning, showing seasonal variations and long-term growth patterns in Brazil, 2018–2023.**
(TIF)

**S1 Table. National impact of the COVID-19 pandemic on suicide attempt notifications among adolescents.**
(DOCX)

**S2 Table. Regional impact of the COVID-19 pandemic on suicide attempt notifications among adolescents.**
(DOCX)

## Author contributions

**Conceptualization:** Juliana D. P. Bulhões, Beatriz Barreto-Duarte, Bruno B. Andrade.

**Data curation:** Gabriela L. Rosier, Klauss Villalva-Serra, Fabio M. H. Sales Filho, Moreno M. Magalhães, Beatriz Barreto-Duarte, Bruno B. Andrade.

**Formal analysis:** Klauss Villalva-Serra, Fabio M. H. Sales Filho, Moreno M. Magalhães, Beatriz Barreto-Duarte, Bruno B. Andrade.

**Funding acquisition:** Beatriz Barreto-Duarte, Bruno B. Andrade.

**Investigation:** Juliana D. P. Bulhões, Gabriela L. Rosier, Klauss Villalva-Serra, Moreno M. Magalhães, Beatriz Barreto-Duarte, Bruno B. Andrade.

**Methodology:** Juliana D. P. Bulhões, Gabriela L. Rosier, Fabio M. H. Sales Filho, Moreno M. Magalhães, Beatriz Barreto-Duarte, Bruno B. Andrade.

**Project administration:** Moreno M. Magalhães, Beatriz Barreto-Duarte, Bruno B. Andrade.

**Resources:** Beatriz Barreto-Duarte, Bruno B. Andrade.

**Software:** Klauss Villalva-Serra, Fabio M. H. Sales Filho, Moreno M. Magalhães, Beatriz Barreto-Duarte, Bruno B. Andrade.

**Supervision:** Moreno M. Magalhães, Beatriz Barreto-Duarte, Bruno B. Andrade.

**Validation:** Juliana D. P. Bulhões, Fabio M. H. Sales Filho, Moreno M. Magalhães, Beatriz Barreto-Duarte, Bruno B. Andrade.

**Visualization:** Gabriela L. Rosier, Klauss Villalva-Serra, Fabio M. H. Sales Filho, Moreno M. Magalhães, Beatriz Barreto-Duarte, Bruno B. Andrade.

**Writing – original draft:** Juliana D. P. Bulhões, Gabriela L. Rosier, Klauss Villalva-Serra, Moreno M. Magalhães, Beatriz Barreto-Duarte, Bruno B. Andrade.

**Writing – review & editing:** Juliana D. P. Bulhões, Gabriela L. Rosier, Klauss Villalva-Serra, Fabio M. H. Sales Filho, Moreno M. Magalhães, Beatriz Barreto-Duarte, Bruno B. Andrade.

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
