## [Decision Letter · Decision Letter 0]

1 Oct 2025

PGPH-D-25-02039

Adolescent Suicide Attempts in Brazil and Impact of COVID-19 Pandemic: A Temporal Analysis

Dear Dr. Barreto-Duarte,

Thank you for submitting your manuscript to PLOS Global Public Health. After careful consideration, we feel that it has merit but does not fully meet PLOS Global Public Health’s publication criteria as it currently stands. Therefore, we invite you to submit a revised version of the manuscript that addresses the points raised during the review process.

This is a well written paper on an important topic. The methodology is sound, and the paper as a whole meets our publication criteria. The results are clearly written. In the discussion section, it would be good to have the authors' perspective on some of their important findings such as why they found the suicide rates to be lower during the COVID pandemic.

We look forward to receiving your revised manuscript.

Kind regards,

Susmita Chandramouleeshwaran

Academic Editor

Journal Requirements:

1. Please send a completed 'Competing Interests' statement, including any COIs declared by your co-authors. If you have no competing interests to declare, please state "The authors have declared that no competing interests exist". Otherwise please declare all competing interests beginning with the statement "I have read the journal's policy and the authors of this manuscript have the following competing interests:"

2. In the online submission form, you indicated that The data that support the findings of this study will be available upon reasonable request to the corresponding author of the study.

3. Uploaded as supplementary information.

3. Please provide separate figure files in .tif or .eps format.

4. Some material included in your submission may be copyrighted. According to PLOS’s copyright policy, authors who use figures or other material (e.g., graphics, clipart, maps) from another author or copyright holder must demonstrate or obtain permission to publish this material under the Creative Commons Attribution 4.0 International (CC BY 4.0) License used by PLOS journals. Please closely review the details of PLOS’s copyright requirements here: PLOS Licenses and Copyright. If you need to request permissions from a copyright holder, you may use PLOS's Copyright Content Permission form.

Potential Copyright Issues:

Figure 1: please (a) provide a direct link to the base layer of the map (i.e., the country or region border shape) and ensure this is also included in the figure legend; and (b) provide a link to the terms of use / license information for the base layer image or shapefile. We cannot publish proprietary or copyrighted maps (e.g. Google Maps, Mapquest) and the terms of use for your map base layer must be compatible with our CC-BY 4.0 license.

Reviewers' comments:

Reviewer's Responses to Questions

**Comments to the Author**

1. Does this manuscript meet PLOS Global Public Health’s publication criteria?

Reviewer #1: Yes

2. Has the statistical analysis been performed appropriately and rigorously?

Reviewer #1: Yes

3. Have the authors made all data underlying the findings in their manuscript fully available (please refer to the Data Availability Statement at the start of the manuscript PDF file)?

Reviewer #1: No

4. Is the manuscript presented in an intelligible fashion and written in standard English?

Reviewer #1: Yes

Reviewer #1: The Topic of the article is interesting and the results are fascinating. Authors are requested to revise as per instructions

1. Abstract: Define exogenous poisoning

2. Key words: Use MESH Words up to 10

3. Add a study settings section to methods: with focus on existing national policy, child rights, relevant health programmes, ongoing conflicts, socio political issues

4. Tables have to be structured as per SAMPL guidelines

5. all the recommendations in this paper are very general they should be rewritten specifically for the study area and in line with the existing health system and education department

6. Authors should discuss with local experts and understand how gender in the study area contributes to more suicide attempts- is it child marriage/teen pregnancy? sex abuse or violence or any other factor? Please elaborate.

7. Authors should describe why covid 19 resulted in drop in suicide - is there corporal punishment in schools? domestic violence dropped? The authors should describe this in the discussion.

8. Authors should connect how at each stage of adolescence, adolescents are prone for mental health risks and suicides and mention the web of causation specific to the study area.

9. Methods and results should be rewritten in a simple manner keeping in mind the amateur readers.

**Do you want your identity to be public for this peer review?** For information about this choice, including consent withdrawal, please see our Privacy Policy

Reviewer #1: **Yes:** Nancy Angeline Gnanaselvam

---

## [Editor Report · Decision Letter 1]

28 Dec 2025

Adolescent Suicide Attempts in Brazil and Impact of COVID-19 Pandemic: A Temporal Analysis

PGPH-D-25-02039R1

Dear Drt Barreto-Duarte,

We are pleased to inform you that your manuscript 'Adolescent Suicide Attempts in Brazil and Impact of COVID-19 Pandemic: A Temporal Analysis' has been provisionally accepted for publication in PLOS Global Public Health.

Best regards,

Susmita Chandramouleeshwaran

Academic Editor